# Manufacturing of Pure Iron by Cold Rolling and Investigation for Application in Magnetic Flux Shielding

**DOI:** 10.3390/ma15072630

**Published:** 2022-04-02

**Authors:** Nitin Satpute, Prakash Dhoka, Marek Iwaniec, Siddharth Jabade, Pankaj Karande

**Affiliations:** 1Department of Mechanical Engineering, Faculty of Science and Technology, Vishwakarma University, Pune 411048, India; pkd@imp-india.com (P.D.); Siddharth.jabade@vupune.ac.in (S.J.); 2Faculty of Electrical Engineering, Automatics, Computer Science and Biomedical Engineering, AGH University of Science and Technology, 30-059 Kraków, Poland; iwaniec@agh.edu.pl; 3Industrial Metal Powder Pvt. Ltd., Pune 412216, India; pankajkarande95@gmail.com

**Keywords:** magnetic analysis, high permeable material, finite element analysis, shielding effectiveness, magnetic shielding

## Abstract

The presented work investigates a novel method to manufacture 98.8% pure iron strips having high permeability and better saturation flux density for application in magnetic flux shielding. The proposed method uses electro-deposition and cold rolling along with intermediate annealing in a controlled environment to manufacture 0.05–0.5 mm thick pure iron strips. The presented approach is inexpensive, has better control over scaling/oxidation and requires low energy than that of the conventional methods of pure iron manufacturing by pyrometallurgical methods. Important magnetic and mechanical properties of the pure iron are investigated in the context of the application of the material in magnetic shielding. Magnetic properties of the material are investigated by following IEC60404-4 standard and toroidal coil test to determine hysteresis curve, magnetic permeability and core losses. The microstructure is investigated with an optical microscope and scanning electron microscopy to study grain size and defects after cold rolling and annealing. The properties derived from the experimental methods are used in finite element analysis to study the application of the material for static, low-frequency and high-frequency magnetic shielding. Theoretical simulation results for magnetic shielding around a current-carrying conductor and micro-electromechanical inductive sensor system are discussed. Further shielding performance of the material is compared with that of the other candidate materials, including that of Mu-metal and electrical steel. It is demonstrated that the pure iron strips manufactured in the present study can be used for magnetic shielding in the case of low-frequency applications. In the case of high-frequency applications, a conducting layer can be combined to ensure the required shielding effectiveness in the case of Class 2 applications.

## 1. Introduction

High-purity iron foils are investigated for various applications, including magnetic shielding and biodegradable medical implants. It is characterized by high magnetic permeability, low coercivity, high saturation flux density, better electrical conductivity, and good mechanical strength. In recent years, pure iron has been investigated for applications in magnetic shielding of electronic devices, current-carrying cable, and healthcare devices [1,2].

Technically pure iron and pure electrolyte iron are the two representations of pure iron. The technically pure iron has 99.6–99.8% purity and is conventionally manufactured with pyrometallurgy. This process involves reducing iron ore in a controlled environment furnace followed by multi-stage purification [3,4]. The basic oxygen process used in the purification during pyrometallurgical methods involves the injection of high-purity oxygen to remove carbon and other impurities [5]. Recently hybrid and combined hydropyrometallurgical methods are reported to overcome several limitations of the conventional pyrometallurgy methods, including higher energy consumption [6]. The hybrid pyrometallurgical approach proposed by Mizuno et al. uses the microwave technique for rapid and efficient heating [7]. The combined hydropyrometallurgical process uses acid or alkaline leaching agents to leach the desired element, and the process is performed at room temperature [8]. Electrolyte iron has impurities lower than 600 ppm and requires controlled environment melting and multistage purification processes [9]. Torrent et al. performed electron beam melting to manufacture electrolytic pure iron having 99.99% purity [10]. Chikoshi et al. used anion exchange and plasma arc melting to remove metallic impurities. The hydrogen reduction technique is used to remove nonmetallic impurities in the pure iron manufacturing process [11].

Traditionally, the electro-deposition and cold rolling approach is used to manufacture copper, aluminum, and silver foils [12]. However, attempts have been made to use electro-deposition and cold rolling methods to manufacture thin pure iron foils of a few hundred micron thicknesses. Recently, this approach has been investigated for pure iron manufacture from high-purity ferrous slat solutions and cold rolling with recrystallization annealing. Majidzade et al. studied the possibility of iron deposition from various aqueous electrolyte solutions [13]. Operating parameters for ethylene glycol solution are investigated to ensure smooth and compact iron deposition. Giridhar et al. investigated the possibility of manufacturing magnetic material from the electro-deposition process [14]. Moravej et al. investigated electro-deposition of iron on titanium electrodes for degradable stent application [15]. The current density was found to be an influencing factor for the strength of the texture, and recrystallization annealing was carried out at 550 °C. Cold rolling process can be used for finishing the initial deposits obtained after the electrodeposition process and thick ingots of pure iron and electrical steel to the final thickness [16,17]. Ma et al. proposed multiple cold rolling passes with intermediate annealing to manufacture metal strips of a ductile material [18]. The parameters were controlled closely to reduce thickness by 15% in each cold rolling pass to eliminate the possibility of cracks during working. The electrolytic process involves the use of electrons as the reduction agents and is used for metal extraction and refining processes [19]. In applying the electrolytic process, Wang et al. investigated the removal of copper and aluminum impurities by solid-state electrotransport phenomenon [20].

The magnetic field created around a current-carrying conductor is vital in the context of human health and electromagnetic interference in electronic/electrical devices. Various standards recommended maximum permissible magnetic field intensity, which is 20 µT for 2 kV/m conductor. Furthermore, international guidelines enforce maximum field intensity of 0.8 µT at 50 Hz frequency in the context of human health [21,22]. Due to high permeability, Mu-metal and electrical steel (Fe-Si alloy) are used as flux shunting materials. To develop an economical substitute for the shielding material, it is necessary to investigate the application of the pure iron strips (manufactured with electrodeposition and cold rolling) for magnetic shielding.

The passive magnetic shielding approach involves the use of flux shunting material and eddy currents cancellation techniques [23]. The flux shunting method involves the use of high permeable material to divert the flux away from the shielded region and is preferred for low-frequency applications. Cazacu et al. evaluated the shielding effectiveness of a flux shunting material around a current-carrying conductor [24]. In the case of high-frequency applications (frequency > 60 Hz), a conducting layer of aluminum or copper is used along with the permeable material for magnetic field shielding [25]. Zeng et al. proposed ferrite coils and an aluminum layer for magnetic shielding to increase the power transmission efficiency of an electro-magnetic system [26].

In order to evaluate the material for magnetic shielding application using the finite element analysis (FEA) approach, it is necessary to perform characterization of the material to determine hysteresis and initial magnetizations curves. Magnetometric and inductive methods are used for the characterization of magnetic materials. The magnetometric methods, including vibrating sample magnetometer (VSM) test, are not suitable for the determination of the hysteresis and initial magnetization curves of high permeable materials such as pure iron and electrical steel, due to the demagnetizing effect on the test specimen [27]. On the other hand, inductive methods as per IEC60404-4 are preferred for the determination of the hysteresis loop and initial magnetization curve of high permeable soft magnetic materials, including that of the pure iron. International standard IEC 60404-4 specifies methods for characterization of high permeable materials to experimentally determine these properties. The standard includes recommended procedures for determining the materials’ initial magnetization and hysteresis curves.

The literature study identifies that there is difficulty in reducing impurities to obtain high-purity iron with use of the conventional and modified pyrometallurgical methods. To avoid the detrimental effects of excess oxygen, it is necessary to use vacuum control for the basic oxygen process during pyrometallurgical methods of pure iron manufacturing [9]. The use of a melting process during the manufacturing of the pure iron carries the risk of significant iron loss due to high vapor pressure [28]. There is difficulty in removing impurities by melting the iron, since few impurities have lower boiling point, which leads to the risk of damage to the furnace lining and a reduction in production [29]. In the case of pure iron manufacturing, there is difficulty in controlling the impurities, since the single purification method is not applicable to all the impurities. Further use of high temperature increases oxidation of the iron, which is difficult to control with the addition of alloying elements including carbon, aluminum and silicon, since these are themselves impurities. There is a challenge in controlling oxidation and de-oxidation elements simultaneously. The above-mentioned methods increase the cost and energy demand of the pure iron manufacturing process. There is a need for an alternate economical method for manufacturing pure iron strips, particularly for application in magnetic shielding.

Although a few initial studies are reported in the references for manufacturing pure iron by electrodeposition and cold rolling, there is no commercial application of this method. It is necessary to further investigate this approach to establish the possibility of large-scale manufacturing of pure iron. It is reported in the literature that pure iron has high magnetic permeability and is an economical material for magnetic flux shielding [30]. Therefore, there is a need to investigate the properties of the material manufactured using the proposed approach in the context of the potential magnetic shielding applications.

In this paper, a new method for manufacturing of 99.8% pure iron strips for application in magnetic shielding is discussed. The presented method is economical and suitable for large-scale production, with better control over scaling/oxidation. The pure iron material is investigated as per IEC60404-4 standard and toroidal coil test to determine important magnetic properties. Scanning electron microscopy and spark emission spectroscopy are used to study microstructure and chemical composition. Further, the properties determined with the experimental methods are used in FEA to calculate shielding effectiveness achieved with the pure iron magnetic shield. The shielding performance of the material under study is compared with that of the other candidate materials. It is demonstrated that the proposed cost-effective solution can be applied for static, low-frequency and high-frequency applications.

The paper is organized as follows: Section 2 explains the manufacturing of the pure iron strips with details of electro-deposition, cold rolling and controlled environment annealing. The necessary material properties of the iron strips, including B-H curve, maximum relative magnetic permeability and eddy currents losses are discussed in Section 3. Investigation of material properties and microstructure is reported in Section 4. A brief discussion on magnetic and material properties is presented in Section 5. Details of the FEA used to estimate shielding effectiveness are discussed in Section 6. Section 7 includes a discussion on magnetic and material properties investigation, simulations for shielding effectiveness, influencing parameters and methods for improvement in magnetic permeability.

## 2. Manufacturing of the Pure Iron Strips

The pure iron strips are manufactured using the process of electro-deposition, which is followed by cold rolling and intermediate annealing cycles. Further details of the process are as follows.

### 2.1. Electro-Deposition of Iron

Electro-chemical deposition from ferrous chloride solution is used to obtain an initial 3 mm-thick pure iron deposit, with detailed process parameters and equipment as given in Table 1.

The deposits were removed from the electrode after the electro-deposition process in 210 mm × 250 mm × 3 mm sheets. The deposit had an inferior surface finish and non-uniform thickness, due to its inherent porous nature. The electrodeposited specimen’s chemical composition is determined using spark emission spectroscopy measurement (Make: Thermofisher Scientific Spark Spectrometer, Pune, India, Model: ARL 3460) and reported in Table 2. It can be noted that the electrodeposited specimen has a negligible presence of impurities and the derived iron deposits have 99.8% purity. The specimen obtained from the deposition process has further been used in cold rolling and annealing cycles as discussed further.

### 2.2. Cold Rolling and Annealing

The electro-deposited specimen was annealed at 1000 °C in an argon environment for 30 min and cooled in the furnace (time taken for cooling from annealing temperature: 16 h). Later, cold rolling was performed with the controlled environment annealing after every rolling pass (Annealing temperature 1000 °C, holding time: 30 min., Argon environment, furnace cooling, time taken for cooling from annealing temperature: 16 h). During each rolling pass, the thickness was reduced by 20–25%, followed by the annealing. The process has been used to reduce the foil thickness from the initial deposit to the final thickness of 0.3 to 0.1 mm.

Detailed specifications and process parameters of the rolling operation are reported in Table 3.

Details of changes in physical properties before and after cold rolling are given in Table 4. It was observed that intermediate annealing is necessary to ensure the absence of cracks and other defects during rolling passes.

## 3. Characterisation of Magnetic Properties

Experimental methods are used to measure the properties with different magnetization conditions. The properties that are investigated include maximum relative magnetic permeability, B-H curve and core losses. Further, the derived magnetic properties are used in FEA to validate the application of the pure iron material for magnetic shielding.

### 3.1. Initial Magnetization and Hysteresis Curve

The magnetic properties of the pure iron strips manufactured in the present study are characterized by following the international standard, IEC 60404-4, which is applicable for the measurement of DC magnetic properties of iron and steel.

The experimental test facility at Veer Electronics Pvt. Ltd., Gandhinagar, Gujrat, India, is used for the analysis (Make: Veer Electronics, Gandhinagar, Gujrat, India, Model: VCT160101). The test facility has maximum limit of maximum applied field strength limit at 10,000 A/m, whereas it can measure flux density up to 2.0 T. Accuracy of the measurement facility is 0.5%. The ring method is used to determine the initial magnetization curve and the permeameter method is used to plot the hysteresis curve of the specimen. Further, details of the magnetic characterization are as follows:A.Sample preparation

Specimen for determination of the initial magnetic characterization curve is prepared from rings of outer diameter 95 mm and inner diameter of 86 mm. Several rings of thickness 0.5 mm each are stacked together in the 12 mm-thick specimen. The rings are carefully prepared by using water jet machining to ensure that magnetic properties are not affected during specimen preparation. The test specimen is wound with copper coils (number of turns is 90) to apply the magnetic field, which has voltage-controlled current generator to vary the magnetic field. Another coil (number of turns is 90) wound on the specimen detects voltage, which is used to estimate magnetic flux density. Values of magnetic field strength and flux density are used to determine the relative permeability of the specimen.

B.
Initial magnetization curve 

The point-by-point method specified in IEC 40604-4 is used to determine the initial magnetization curve. During the application of the magnetic field at each point, 30 s are given to ensure that the magnetic field is stabilized at each of the point. The maximum amplitude is varied up to 10,000 A/m and voltage in the secondary coil is measured to calculate magnetic flux density in the specimen. Figure 1 shows variation in the flux density with the applied magnetic field. Further variation in the relative permeability is given in Figure 2. Results shown in Figure 1 and Figure 2 indicate the pure iron specimen’s expected performance, which reveals maximum relative permeability of 7818. Moreover, Figure 1 does not indicate saturation of the pure iron, which is because the saturation flux density of pure iron is much higher than the upper measurement limit in the test facility.

C.
Hysteresis loop 

A hysteresis loop for the pure iron is plotted with continuous recording using the permeameter method. According to the requirements of the IEC60404-4 standard, the current value necessary to produce the required magnetic field strength of 600 A/m was passed through the primary coil. The current was slowly varied between negative to positive maximum values, where the cycle was completed in 60 s and a time duration of 20 s was provided to ensure stabilization of the magnetic field in the specimen. The magnetic field strength values are plotted against the flux density in relation to zero points of both parameters.

The hysteresis loop for the pure iron specimen is shown in Figure 3. Important parameters determined from the initial magnetization curve and hysteresis loop are reported in Table 5.

The observations for the non-linear nature of B-H curve and magnetic properties reported in Table 5 are in close agreement with the magnetic properties for pure iron (which is manufactured from iron powder) reported in the literature [31,32,33,34]. However, it can be noted that the procedure followed for fabrication of the pure iron sheets in this work (i.e., electro-deposition and cold rolling) is different to that of the fabrication procedures reported in the references.

### 3.2. Investigation of Core Loss

Core loss is an important property of a magnetic material, and it occurs due to alternate magnetization. The core loss is the cumulative effect of eddy current and hysteresis losses, whereas lower core loss is desired in application in electrical machines, including electric motors and transformers. The classical two-winding method with toroidal shaped metal strips and coil windings is used to characterize the core losses. The experimental facility at Department of Mechanical Engineering, Charusat University, Gujrat, India, is used for the experimentation. The arrangement uses toroidal shaped iron foils (outer diameter: 35 mm, inner diameter: 20 mm, thickness: 0.16 mm), primary and secondary windings (number of turns is 10 for each) are used during the analysis. Digital multimeter (make: Iwatsu, Tokyo, Japan, model: VOAC7602) and digital oscilloscope (model: DSOX3000) are used for measurement of current and voltage, which are further used to derive the core losses. Measurement of the core loss is performed at 50 Hz, and variation of the losses along the flux density is shown in Figure 4. It can be seen that the maximum core loss of 8.1 W/kg is observed at flux density of 1.7 T. The core loss for the iron strip manufactured in the present work is compared with that of the literature (where the iron foils are manufactured from pure iron powder and continuous casting with hot rolling) in Table 6. It can be noted that the observations for the core loss in the present work are in agreement with those of the loss values reported in the literature. The variation in the core loss in the present work and literature (as seen from Table 6) is due to the difference in the density and resistivity of the material manufactured with different methods. The core losses can be further reduced by using ceramic coating on the iron foils and grain size refinement to about 250–300 µm. However, in the case of conventional (Fe-Si) electrical steel, the core loss is much lower (less than 0.5 W/kg), which makes this material more suitable for electrical motors, transformers and other applications that have alternating current fields [35].

Core loss values for pure iron manufactured using the other conventional methods are reported from the literature in Table 6. It can be noted that the core loss values greatly vary with the thickness of the iron strips.

## 4. Microstructure and Chemical Composition Analysis

### 4.1. Energy Dispersive X-ray Spectroscopy (EDS) Analysis and Scanning Electron Microscope (SEM) Analysis

EDS and SEM are performed to reveal the pure iron strips’ microstructure details and chemical composition with a test facility available at College of Engineering, Pune, India. (Make: Zeiss, Aalen, Germany, Model: Sigma HV). The investigation is conducted at various stages of manufacturing for 0.3 mm-thick pure iron strip as follows:i.After completion of the last cold rolling pass and before annealing;ii.After completion of the last cold rolling pass and after argon environment annealing;iii.After completion of the last cold rolling pass and after hydrogen environment annealing.

The microstructure of the specimen which has completed the last pass of cold rolling and before annealing is shown in Figure 5a. It can be noted that the microstructure has defects and microscopic cracks which are introduced due to cold rolling. The microstructure of the argon annealed pure iron strip (details of argon annealing are discussed in Section 2.2) is shown in Figure 6a, whereas EDS analysis is shown in Figure 6b. The microstructure of hydrogen-annealed iron strips (at 980 °C for 30 min, cooling from 980 °C to room temperature in 30 s, details discussed further in Section 7) is shown in Figure 7a and EDS results shown in Figure 7b. It can be noted from Figure 6a and Figure 7a that cracks, inclusions, and other defects are not present in the microstructure after annealing operations. Further, EDS results in Figure 5b, Figure 6b and Figure 7b ensure the purity of iron, which does not reveal the presence of oxygen and other elements.

### 4.2. Microstructure Analysis with Optical Microscope

The microstructure of the pure iron material during various phases of manufacturing is shown in Figure 8, as observed with an optical microscope of 100× magnification. It can be seen that the electro-deposited specimen exhibits a large particle size up to 200 µm, which changes to relatively coarse grains (of size 2–3 mm) after annealing operation. The specimens subjected to rolling operation show elongated grains, which change to randomly oriented non-uniform grains of size 70–80 µm post annealing operation. The microstructure shows pure and homogeneous composition, which is due to slow cooling during annealing, and negligible presence of impurities.

The four-point probe method is used for the measurement of electrical resistivity of the pure iron strips, which was found to be 9.65 × 10^−8^ Ω-m.

## 5. Conclusions from Magnetic and Material Characterization

The 99.8% pure iron foils fabricated with electro-deposition and cold rolling have better permeability (95.4% higher, the maximum relative permeability of electrical steel: 4000 and maximum relative permeability of the pure iron: 7818) than that of conventional Fe-Si electrical steel (Grade: ASTM 30F677). The observation for better magnetic permeability of the pure iron is in agreement with the earlier results reported in the references [38,39,40]. Further, the saturation magnetic flux density of the iron foils is better than electrical steel. However, core loss is higher for the pure iron strips than that of the electrical steel, which is attributed to the reduced resistivity of the pure iron compared to electrical steel.

Permeability of Mu-metal (ASTM A-753 Alloy 4, Maximum permeability: 350,000) is higher than that of the pure iron strips. However, the pure iron strips have higher saturation flux density than Mu-metal and electrical steel (saturation flux density of Mu-metal is 0.6 T). This observation is in consonance with earlier findings reported in the literature [38,40,41]. Therefore, the use of pure iron strips can be investigated for certain applications of magnetic flux shielding.

Investigation with SEM reveals the absence of cracks and inclusions in the pure iron strips after annealing. Further, EDS analysis indicates that the sample contains pure iron and negligible impurities.

It can be noted that the pure iron is a promising low-cost alternative to electrical steel for magnetic shielding. Accordingly, a detailed analysis of flux shunting and eddy currents cancellation approaches for the pure iron strips are investigated by using FEA in the further section. The material and magnetic properties derived from the experimentation, as are discussed in this section, are used in the FEA numerical simulation.

Detailed analysis of the pure iron strips manufactured in the present study is compared with the ASTM 30F677 (hereafter referred to as the electrical steel) and ASTM A-754 Alloy 4 (hereafter referred as the Mu-metal).

## 6. FEA for Magnetic Flux Shunting

Various magnetic, chemical and microstructure properties of the pure iron are investigated in Section 3 and Section 4. It is demonstrated that the pure iron strips possess better permeability and higher saturation flux density, which are important properties for electro-magnetic shielding application. In this section, application of the pure iron strips for electro-magnetic shielding is investigated. A FEA is used to investigate performance parameters when the pure iron strips are applied for static and high frequency shielding applications. Furthermore, shielding performance of the pure iron material is compared with other candidate materials for magnetic shielding which includes electrical (Fe-Si) steel and Mu-metal. In order to ensure better accuracy of the FEA simulation results, material properties derived in Section 3 are used for FEA simulation.

The FEA simulation for determination of magnetic field intensity around a current-carrying conductor is performed in order to compare performance of different shielding materials. The analysis has evaluated two dimensional electro-magnetic models of the current-carrying conductor in ANSYS-18.2 APDL. The geometry in the analysis includes a round shaped copper conductor, surrounding air and shielding material around the conductor. Two types of shunting topologies are considered in the analysis, as shown in Figure 9a,b. Figure 9a illustrates open shunting method where two plates are placed at top and bottom of the conductor at a distance of ‘*g*’ from the conductor (*g* = 150 mm). The closed shunting method shown in Figure 9b has the conductor enclosed by the shielding material with straight and polygon-shaped material. Details of the Ansys elements and material properties defined for each of the components are given in Table 7.

During the FEA simulation, a current is applied by realizing potential difference along the length of the conductor. The diameter of the cable is 25 mm and a current of 800 Amp is passed through the cable. FEA is used to determine the magnetic flux density around the conductor. The dimensions, arrangement, and current in the conductor cables are referred for the applications in under-ground/overhead cable and applicable standards for calculating magnetic flux shielding effectiveness [42,43].

The simulation results with different materials and topologies are discussed in the next section.

## 7. Results and Discussion

Attempts were made to improve the permeability of the pure iron strips manufactured in the present study by annealing the steel in a hydrogen environment followed by rapid cooling to room temperature. Tubular furnace (Make: Therelek furnaces, Maharashtra, India, Glass tube for specimen with inner diameter 55 mm 300 mm heating length, maximum temperature: 1100 °C) is used to carry out the annealing operation. The pure iron strips manufactured for the present study with the thickness of 0.3 mm were heated at 980 °C in the presence of hydrogen for 30 min. Further the specimens were cooled from the annealing temperature to room temperature in 30 s by using a special fixture illustrated in Figure 10. The fixture has a stainless steel container along with provision of a jacket to place liquid nitrogen. This assembly is kept in a box with glass wool to ensure low temperature inside the stainless steel chamber. The annealed pure iron strips were kept inside the chamber immediately after completion of the annealing in the tubular furnace. The specimens were picked from the tubular furnace and placed into the special cooling fixture to rapidly cool the pure iron strips. It was possible to reduce the temperature of the iron strips without any warpage and deformation to room temperature. Later the strips were annealed at 150–200 °C for 20 min to reduce internal stresses induced due to fast cooling. SEM and EDS analysis of the hydrogen-annealed samples are discussed in Section 4.1.

The maximum relative permeability of the pure iron strips manufactured in the present study is estimated by comparison of inductance of the pure iron strips with that of the material with known maximum relative permeability (i.e., Fe-Si electrical steel). Schematic arrangement in the experimental set up is illustrated in Figure 11, which has the test specimen (stack of 0.3 mm thick, 5 strips of hydrogen-annealed pure iron with insulating Teflon layers in between) and reference specimen with known maximum relative permeability (stack of the electrical steel, 0.3 mm thick, five strips sandwiched with Teflon layers in between). The copper coils (50 turns of 28 gauge wire) are wound on the reference and test specimen. Further, inductance of both the coils is measured with Inductor Capacitor Resistor (LCR) meter (Make: Metravi 4070D West Bengal, India, Range: 200 µH–2000 H). The maximum relative permeability of the hydrogen-annealed pure iron strips is calculated by using the following equation:(1)$$\mu_{t}=\mu_{r}\frac{L_{t}}{L_{r}}$$
where

*µ_r_*: relative permeability of reference specimen*L_t_*: inductance of the test specimen*L_r_*: inductance of the reference specimen

**Figure 11 materials-15-02630-f011:**
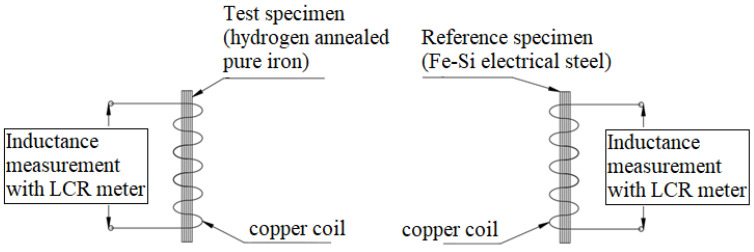
Schematic arrangement in the test set up to estimate relative permeability.

The maximum relative permeability of the test specimen (hydrogen-annealed pure iron strips) estimated from the above analysis is 62,000, which indicates that hydrogen annealing and rapid cooling ensures higher permeability. However, the other properties of the hydrogen-annealed pure iron including core losses, hardness, and mechanical strength, are still to be investigated. The experimental results for better magnetic relative permeability of the hydrogen-annealed specimen are in consonance with earlier findings reported in the literature [35,44,45,46].

The pure iron strips manufactured in the present study are investigated for magnetic shielding application. The study involves the investigation of magnetic flux density around a current-carrying conductor with and without the use of shielding material. Further, the shielding effectiveness results are compared with that of the other candidate materials including the Mu-metal (ASTM specification: A-753 Alloy 4) and the electrical steel (ASTM specification: 30F677) in case of static and high frequency applications. Shielding effectiveness is calculated based on the magnetic flux density at distance of 1 m, around a conductor carrying a current of 800 Amp (Ates et al., 2021), with and without the use of the shielding material *(Shielding effectiveness = 20 log_10_ (|B_o_|/|B_s_|), B_o_ is the magnetic flux density at 1.0 m without use of shielding material and B_s_ is the flux density at 1.0 m with use of the shielding material)* [47].

Initially, shielding effectiveness is calculated for open shielding topology illustrated in Figure 9a. Finite element simulations are performed with three different materials, including the pure iron, the Mu-metal and the electrical steel, to determine shielding effectiveness with various values of the shield thickness. During the simulations, values of the parameters ‘*l*’ and ‘*g*’ are selected according to the space available around an overhead or underground cable (*l* = 200 mm, *g* = 150 mm). Simulation results indicate that maximum shielding effectiveness can be achieved with thickness of 5 mm, 4.2 mm and 3.2 mm for Mu-metal, electrical steel, and the pure iron, respectively. It can be noted that the lower saturation flux density of Mu-metal necessitates higher shield thickness to ensure better shielding effectiveness. Further maximum values of the shielding effectiveness that can be achieved with use of the Mu-metal, the electrical steel and the pure iron strips are 0.291, 0.263 and 0.275, respectively.

Finite element simulation indicates that the closed shunting topology illustrated in Figure 9b ensures higher values of shielding effectiveness than that of the open shunting topology. Simulation results for magnetic flux density in the case of 3.5 mm-thick pure iron strip are shown in Figure 12a and the contour of the magnetic flux lines is shown in Figure 12b. It can be observed that maximum numbers of flux lines are passing through the shielding materials, which results in ensuring significant shielding effectiveness as compared to open topology. FEA simulations performed for the closed topology geometry indicate that dimensions of the shielding geometry including ‘*l*’, ‘*d*’, and ‘*g*’ influence the shielding effectiveness. It is observed that greater values of dimensions ensure better shielding effectiveness. However, the values of these dimensions are selected according to the space available around a conductor in the case of underground and overhead cables (*l* = 200 mm, *d* = 100 mm, *g* = 150 mm). Further, the influence of the shielding material thickness on the shielding effectiveness is investigated and results are shown in Figure 13. It can be noted that the pure iron material ensures shielding effectiveness of 12.21, with the least thickness of 2.4 mm; thereafter, an increase in the thickness does not lead to further improvement in the shielding effectiveness. In case of the Mu-metal and the electrical steel, maximum shielding effectiveness of 12.72 and 11.87 is observed for thickness of 3.0 mm and 3.6 mm, respectively. It can be noted that Mu-metal ensures higher shielding effectiveness due to better permeability. However, it requires higher thickness to reach maximum effectiveness, due to lower saturation flux density. The pure iron delivers better shielding effectiveness with lower thickness than that of the electrical steel, due to the fact that the pure iron has higher permeability and saturation flux density.

The magnetic flux density values at distances of 1 m and 2 m from the current-carrying cable, as are determined with pure iron shielding material for static and low-frequency applications are 39.2 µT and 18.1 µT, which are lower than the recommended limits for safe operation in the context of human health, which have recommended maximum limits of 2 × 10^2/^f^2^ (mT) [48].

In the case of static and low-frequency applications, pure iron manufactured for the present study will have material saving and provide an economic solution (the cost of the pure iron strips is 60% that of the Fe-Si electrical steel and 5% to that of the Mu-metal), although the shielding effectiveness of Mu-metal is marginally higher than that of the pure iron. However, in the case of an application (>1 MHz), flux shielding with eddy currents cancellation along with that of flux shunting are preferred. High-frequency application necessitates the use of a sandwich structure with high permeable material (for flux shunting) and a conducting layer (for eddy currents cancellation), which is investigated further according to the procedure followed in the reference [49].

Experimentation is performed to reveal the effect of plastic deformation on the relative permeability of the pure iron strips. Three types of specimens are used in the two-winding method with toroidal shaped metal strips, having different intensity of plastic deformation. The details of strain induced in the strips and measured maximum relative permeability are reported in Table 8, which shows that the permeability is not affected, due to lower plastic deformation. However, in case of higher plastic deformation, the maximum relative permeability is reduced, which necessitates the application of annealing process to restore the permeability after plastic deformation.

Electromagnetic shielding for low-frequency applications can be achieved by using high permeable flux shunting material. On the other hand, in the case of high-frequency applications (above 1 MHz), a combination of flux shunting material along with an electrically conducting layer for eddy currents cancellation is preferred [27,50]. A finite element simulation is performed to estimate the shielding effectiveness with the use of the pure iron material (manufactured in the present study) along with two types of conducting material (aluminum and copper with 1 mm-thick iron and conducting material). Figure 14 illustrates the geometry where the conducting layer is placed outside the flux shunting pure iron material along with the source of magnetic source in the form of current-carrying conductor placed at the center. High-frequency electromagnetic analysis is performed in Ansys 18.2 with the geometry illustrated in Figure 14. The maximum value of magnetic flux density is determined at a distance of 1 m from the source which is further used to estimate shielding effectiveness when an alternating magnetic field of the desired frequency is created. The FEA results indicate that the shielding effectiveness achieved with the use of the aluminum and copper layer along with the ferromagnetic flux shunting material is 36.6 and 40.36, respectively. The higher shielding effectiveness with copper material is on account of the lower electrical resistivity than that of the aluminum material. It can be concluded that the sandwich structure of the pure iron and conducting copper layer can ensure a shielding effectiveness in the range 20–40, which is suitable for Class 2 applications in magnetic shielding.

The pure iron manufactured in the present study has a very small presence of impurities as in Table 2 and Figure 6, Figure 7 and Figure 8. Further, it is possible to reduce the oxygen level from the present value of 105 ppm to about 5 ppm by using hydrogen annealing after completion of all the cold rolling passes. Moreover, it is demonstrated that the present method can achieve good control over impurities, oxidation, inclusions, cracks, and other defects. Investigation of the material properties reveals that the pure iron can ensure very high relative magnetic permeability and saturation flux density. The pure iron has higher core losses and there is scope to improve the core losses with the incorporation of controlled impurity by doping technique. Further, an optimum balance can be ensured between the doping content and hydrogen annealing to ensure a trade-off between permeability and core losses.

Application of low-frequency magnetic shielding of the pure iron foils is proposed for a miniaturized linear variable differential transformer (LVDT). The objective of the study is to propose the pure iron magnetic shield to ensure that the maximum error of the sensor is limited to 2% over the entire working stroke limit when the LVDT is subjected to an external magnetization field of 800 A/m placed at the distance of 50 mm. The magnitude of the external magnetic field is selected based on 1 mT flux density in the air, which corresponds to the value of the interfering magnetic field encountered by LVDT [51,52]. The distance of the external magnetic field from the LVDT corresponds to space available in an industry and research application of the miniaturized LVDT. Further, the shielding performance achieved with the pure iron is compared with that of the Mu-metal and electrical steel.

LVDT is used as a position sensor and comprises three copper coils and one ferromagnetic core. The arrangement and dimensions of the miniaturized LVDT are shown in Figure 15. Important specifications are given in Table 9, which are chosen based on the commercially available sensor [53]. The primary coil is placed at the center, and two secondary coils are placed on both sides of the primary coil. Input AC voltage is provided at the primary coil, generating a magnetic field along the ferromagnetic core. When the core is at the central position, both the secondary coils share equal magnetic flux and generate equal voltage. Further, as the core is displaced from the mean position, the magnetic flux across the secondary coils changes, which results in unequal voltages across the two secondary coils. The secondary coils are used in a differential arrangement, such that, the differential voltage output is the subtraction of the voltages in the two secondary coils. The differential voltage output varies linearly with the core displacement and is used to determine the core displacement. Accuracy of the LVDT is a critical issue when the sensor is located close to a source of an external magnetic field (an electromagnet), since the magnetic field of the electromagnet interferes with the LVDT magnetic field to introduce error in the measurement.

Design of a magnetic shield is proposed for the LVDT in the form of 0.3 mm-thick pure iron foil placed with 5 mm gap between the outer casing, as illustrated in Figure 16. The thickness of the pure iron strip and distance from the LVDT casing are chosen based on a MEMS sensor’s space and for ensuring better magnetic shielding against the external magnetic field. The electromagnet is placed such that the magnetic fields of the sensor and electromagnet are parallel to each other. FEA with ANSYS Maxwell is performed to investigate the influence of the electromagnet presence near the LVDT on voltage output from the secondary coils. Further, the performance of the LVDT is investigated with the incorporation of the pure iron magnetic shield. Model of the LVDT and the electromagnet are created in ANSYS Maxwell and are shown in Figure 17a, whereas the electrical circuit of LVDT is shown in Figure 17b.

Details of properties of all the components defined in ANSYS are reported in Table 10. During the FEA simulations core of the LVDT is displaced from center to left and right through the stroke limit and differential voltage output from the secondary coils is measured. To evaluate the effect of magnetic shielding in presence of the external magnetic field near the LVDT, FEA simulations are performed for the following three conditions:LVDT operated without presence of electromagnet and pure iron magnetic shield;LVDT operated with the presence of the electromagnet and without the pure iron magnetic shield;LVDT operated with the presence of electromagnet and pure iron magnetic shield.

The FEA simulation results for the variation of the secondary coil differential voltage along displacement of the core, for the above-mentioned three cases, are plotted in Figure 18. It can be observed from Figure 4 that the maximum voltage output from the secondary coils without the presence of the electromagnet is 156.96 mV, which changes to 193.05 mV in the presence of the electromagnet. The maximum error introduced due to the presence of the external magnetic field is 23.2%. Further, the incorporation of the pure iron magnetic shield, when the electromagnet is present near the LVDT, results in the maximum secondary coil voltage of 159.8 mV. Incorporation of the pure iron magnetic shield mitigates the effect of the external magnetic field and reduces the maximum error to 1.81%. Further, FEA simulations are performed to determine the maximum error of LVDT due to the external magnetic field, using Mu-metal and electrical steel (FE-Si) shielding. The simulation results indicate that the maximum error with Mu-metal and electrical steel is 1.2% and 5.71%, respectively. It can be concluded that the Mu-metal shield exhibits only minor reduction in the maximum error than that of the pure iron shield. On the other hand, electrical steel shield results in a higher percentage error than that of the pure iron shield.

Various methods of technically pure iron manufacturing are summarized in Table 11. The presented method for manufacturing of pure iron uses ferrite salts for electrodeposition of pure iron, followed by cold rolling and intermediate annealing. It requires very close control over the composition and pH value of the electrolyte solution to ensure the required iron purity, and it does not involve complex and difficult controls. In the present study, electrodeposition tanks of 1000 L capacity have been used, such that each of the tanks produces 120 kg of pure iron in 96 h. Further, the possibility of mass production of the pure iron is confirmed by completing 120 numbers of cold rolling passes and annealing for 150 sheets (of dimension 100 mm × 100 mm) in one hour. It is possible to use the presented method to manufacture pure iron strips having a few hundred microns of thickness, which are useful in magnetic shielding.

On the other hand, the conventional and modified pyrometallurgical processes reported in references [9,54,55,56] require expensive methods including inert/vacuum environment processing and melting with plasma arc, micro-wave, or electron beam techniques. Further, most of the processes deliver steel ingots, which later need to be hot rolled and cold rolled with intermediate annealing in order to manufacture the pure iron in the form of thin strips. Use of the expensive and complex control during these conventional processes results in very high prices of the thin pure iron strips. The price of the 99.8% pure iron manufactured with the conventional methods for 100 mm × 100 mm size sheet having a thickness of 0.3 mm is USD 68.6 and for 0.05 mm thickness, the price is USD 164.9. On the other hand, the manufacturing cost of the 99.8% pure iron with the proposed method is much lower (0.3 mm-thick sheet of 100 mm × 100 mm dimension at USD 3.98 and 0.05 mm-thick sheet at USD 5.25). The manufacturing cost of the proposed pure iron strips is 5.8% (for 0.3 mm-thick strip) and 3.2% (for 0.05 mm-thick strip) that of the price of the available product in the market. Further, the cost of electrodeposition, heat treatment (controlled environment annealing and tempering), and cold rolling constitute 51%, 39%, and 10% of the total cost, respectively. It can be concluded that the presented method of pure iron manufacturing is inexpensive and does not involve complex and costly process controls.

The method of hydrogen reduction of iron ore and electrolysis of iron oxides reported in the references [57,58,59] results in a purity level which is much lower than that of the present work. Moreover, the methods reported in references [9,57,58,59] represent prototype studies and commercial applications of the methods are still not available for large-scale manufacturing of the pure iron foils.

**Table 11 materials-15-02630-t011:** Methods for manufacturing of technically pure iron.

Reference	Purity	Manufacturing Method
[58,59]	99.6–99.8%	Pure iron ingot manufactured with pyrometallurgical method subjected to number of cold rolling passes and hydrogen annealing
[59]	97.6–97.8%	Vacuum-induction melting of iron ingots, two stages of hot rolling and cold rolling passes with intermediate annealing
[9]	99.89%	Direct reduction and multistage purification for producing pure iron ingots
[59]	72.19%	Hydrogen reduction of iron ore to manufacture pure iron ingots
[59]	99.5%	Electrolysis of iron oxide in alkaline solution with two inert electrodes
[59]	77%	Electrolysis of molten iron oxide with powder form of magnetite, aldrich, agnesia and silica
This work	99.8%	Electrolysis of ferrous chloride, cold rolling and intermediate annealing

The future scope of the study on the investigation of magnetic shielding application of the pure iron material includes the determination of optimum annealing parameters to ensure higher magnetic permeability and the investigation of the influence of the annealing parameters on mechanical properties of the material under study.

## 8. Conclusions

The paper presents a new method for large-scale manufacturing of 99.8% pure iron strips by using electro-deposition, cold rolling, and intermediate annealing in a controlled environment. Important process parameters are discussed to manufacture pure iron strips having a thickness within the range 0.5–0.3 mm. SEM and optical microscopy reveal that the controlled environment annealing has eliminated the defects introduced during cold rolling. The presented method ensures significant cost advantages in manufacturing the pure iron strips and has better control over impurities, scaling, oxidation, and grain size. Detailed characterization is performed as per IEC 60404-4 standard, toroidal test, and other methods to determine the initial magnetization curve, hysteresis loop, and mechanical properties of the material. It is revealed that the pure iron strips have a maximum relative magnetic permeability of 7818 and higher saturation flux density.

The pure iron strips are evaluated for use in static, low-frequency, and high-frequency magnetic shielding applications. Static and low-frequency magnetic shielding has been demonstrated with the pure iron strips as flux shunting material, whereas high-frequency application is evaluated with the sandwich structure of high permeable (flux shunting) and conductive layers (for eddy currents cancellation). A theoretical study with FEA is used to determine the shielding effectiveness achieved with the material, and the shielding effectiveness is compared with that of other candidate materials, including electrical steel and Mu-metal. It is concluded that the pure iron can deliver shielding effectiveness up to 40.36 for high-frequency applications, which is useful for magnetic field shielding in the case of Class 2 applications. In the case of low-frequency magnetic shielding of an inductive sensor with pure iron strips, the maximum error of the MEMS LVDT is reduced from 23.2% to 1.81%.

## Figures and Tables

**Figure 1 materials-15-02630-f001:**
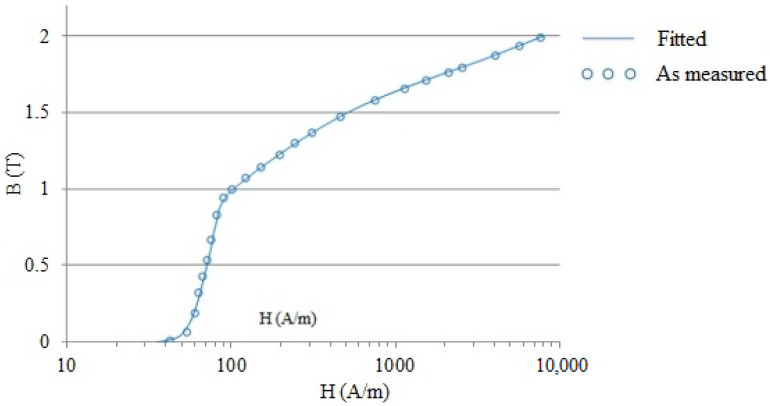
Initial magnetization curve for the pure iron.

**Figure 2 materials-15-02630-f002:**
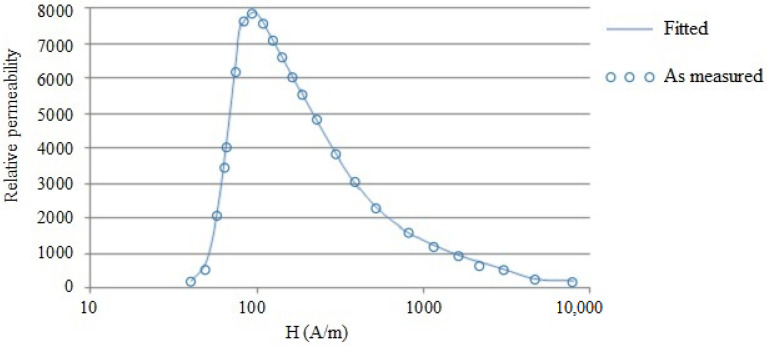
Variation of relative permeability with applied magnetic field.

**Figure 3 materials-15-02630-f003:**
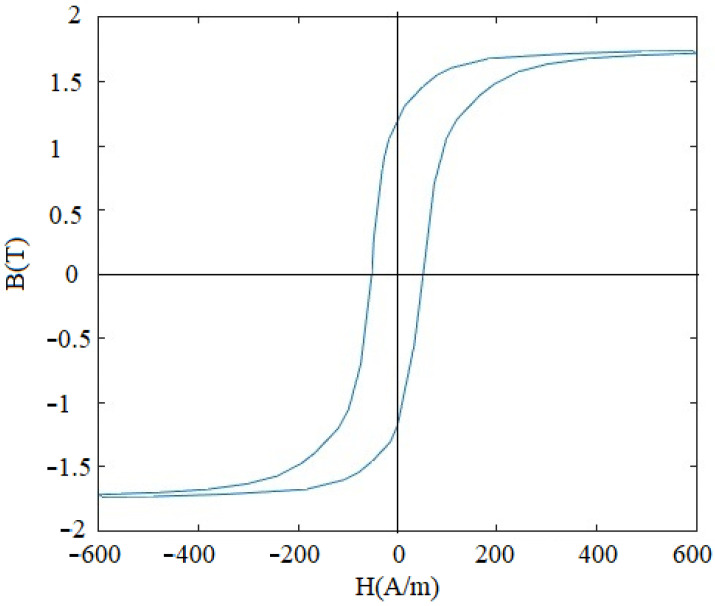
Hysteresis loop for the pure iron.

**Figure 4 materials-15-02630-f004:**
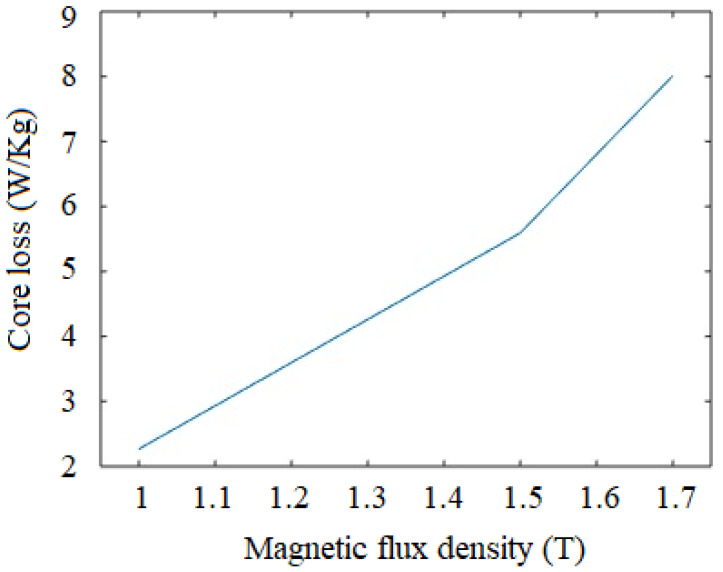
Variation of core losses with applied magnetic flux.

**Figure 5 materials-15-02630-f005:**
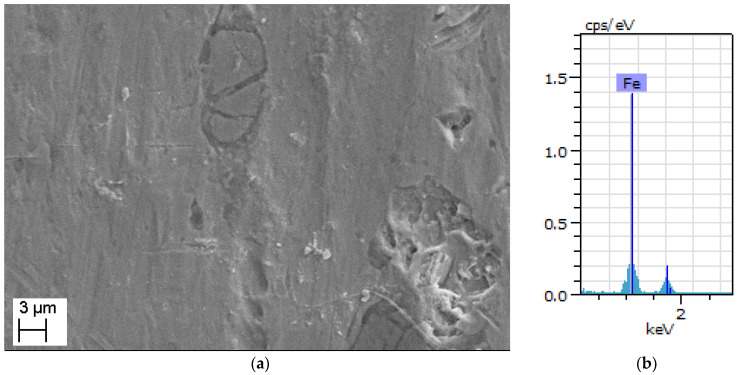
(**a**): SEM image of pure iron after one pass of cold rolling and before annealing. (**b**) EDS analysis of pure iron after one pass of cold rolling and before annealing.

**Figure 6 materials-15-02630-f006:**
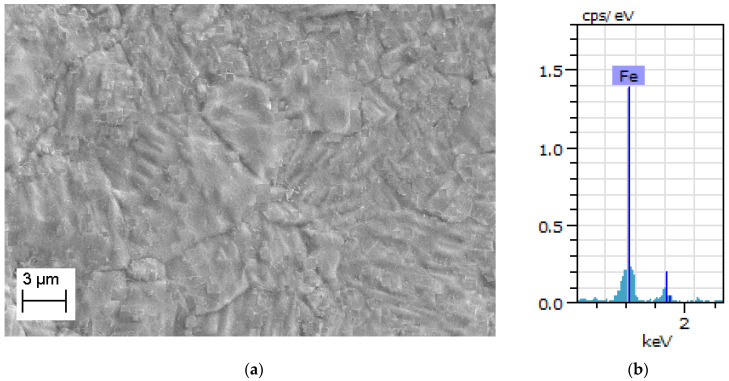
(**a**) SEM image of argon annealed pure iron. (**b**) EDS analysis of argon annealed pure iron.

**Figure 7 materials-15-02630-f007:**
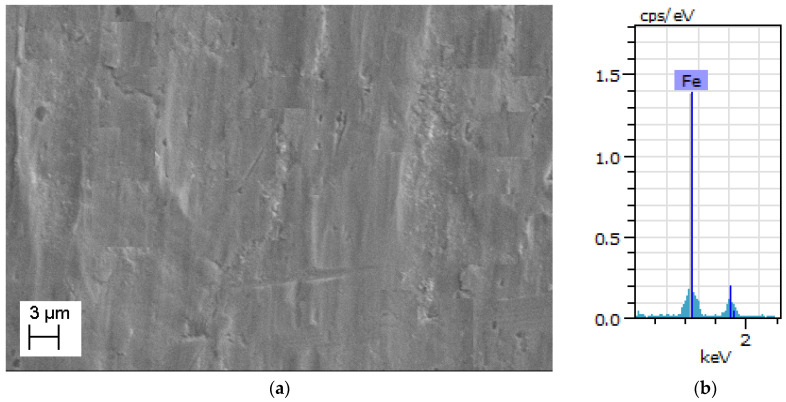
(**a**) SEM image of hydrogen-annealed pure iron. (**b**) EDS analysis of hydrogen-annealed pure iron.

**Figure 8 materials-15-02630-f008:**
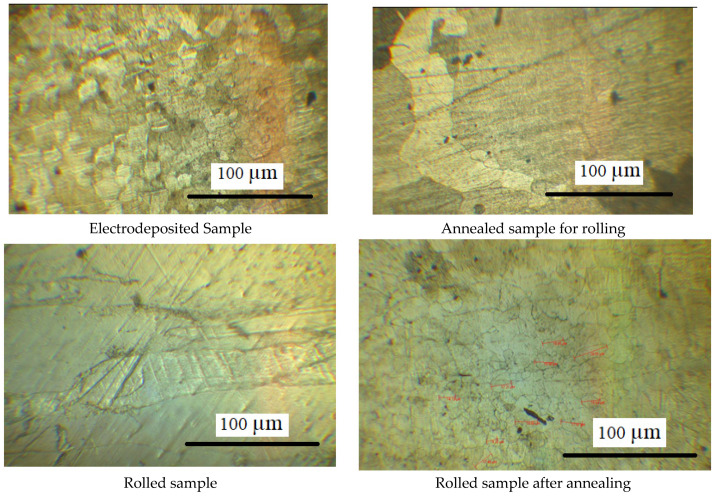
Microstructure of pure iron during various phases of manufacturing.

**Figure 9 materials-15-02630-f009:**
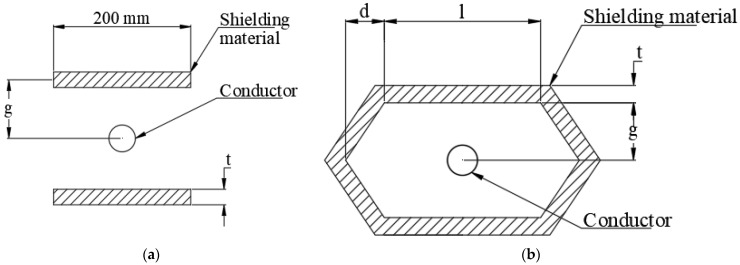
(**a**) Open shunting topology. (**b**) Closed shunting topology.

**Figure 10 materials-15-02630-f010:**
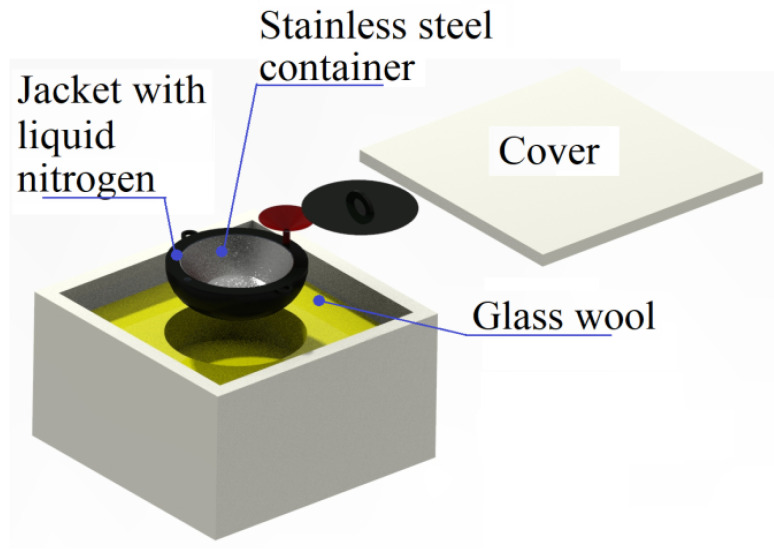
Illustration of the arrangement in the special cooling fixture.

**Figure 12 materials-15-02630-f012:**
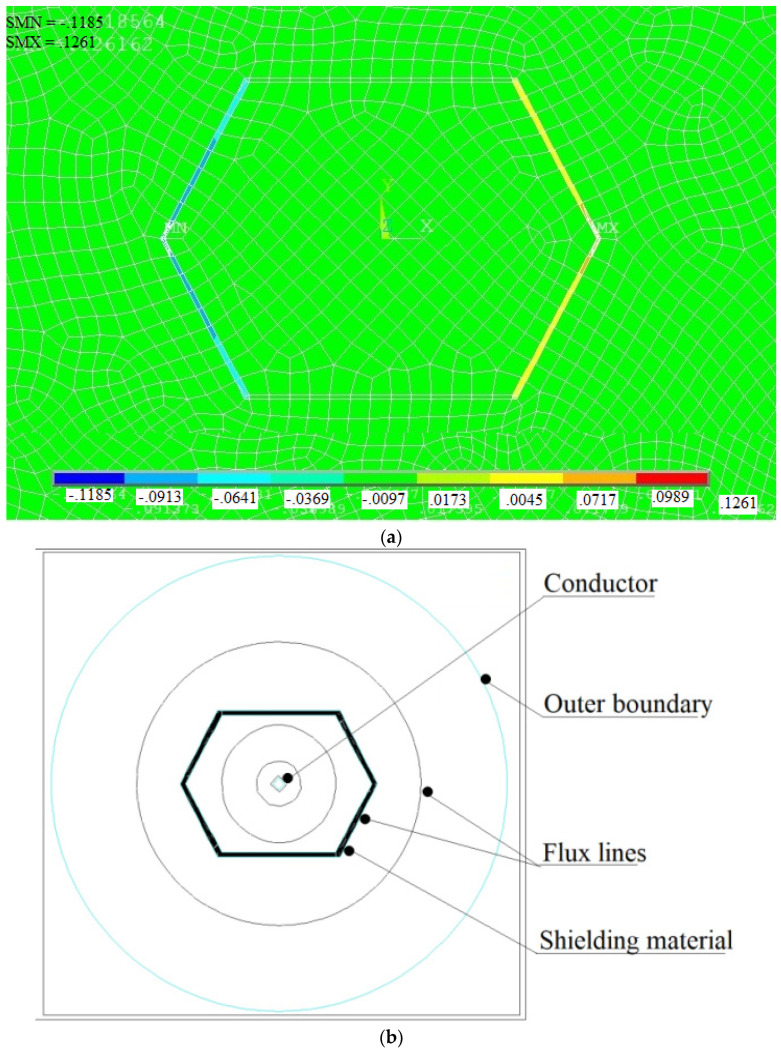
FEA simulation results (**a**) Magnetic flux density with closed topology pure iron shielding of 3 mm thickness. (**b**) Magnetic flux distribution along the shielding material.

**Figure 13 materials-15-02630-f013:**
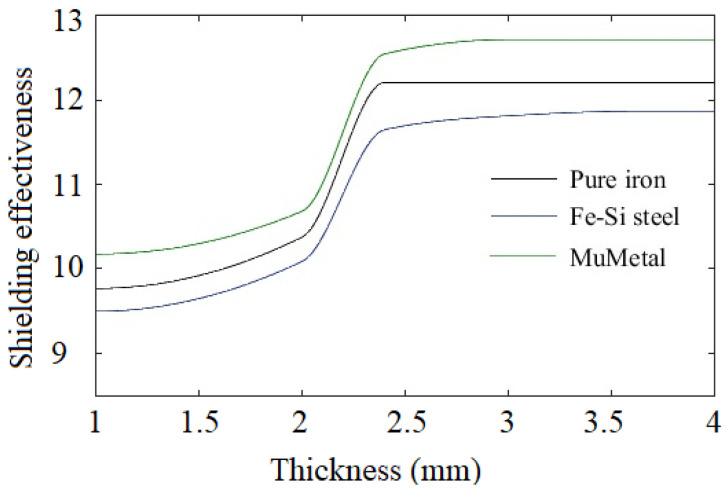
Shielding effectiveness with different shielding material.

**Figure 14 materials-15-02630-f014:**
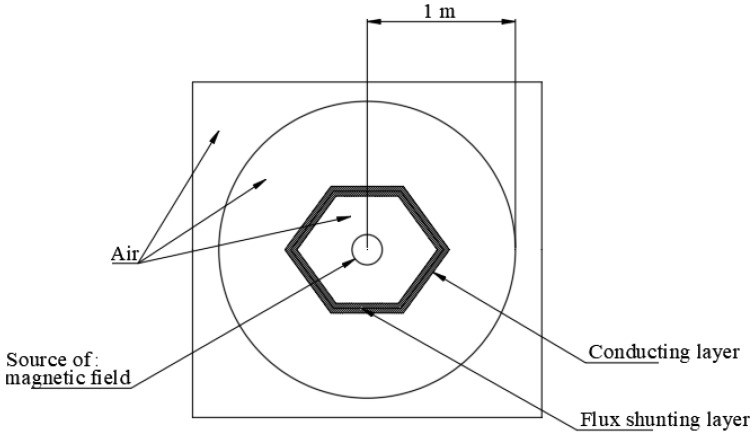
Arrangement of high frequency magnetic field shielding.

**Figure 15 materials-15-02630-f015:**
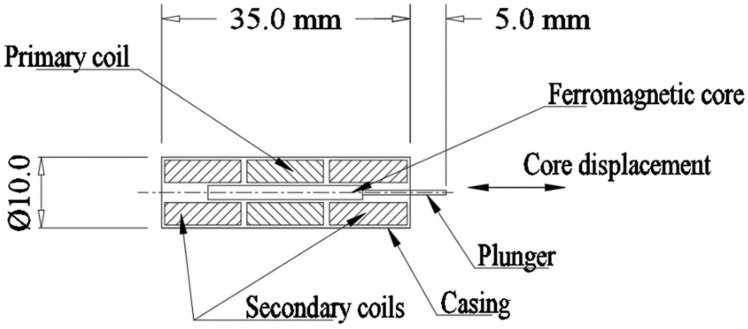
Details of the MEMS LVDT.

**Figure 16 materials-15-02630-f016:**
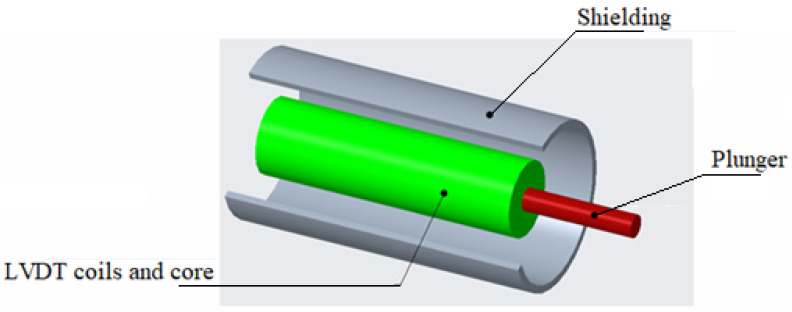
Shieling material around the LVDT.

**Figure 17 materials-15-02630-f017:**
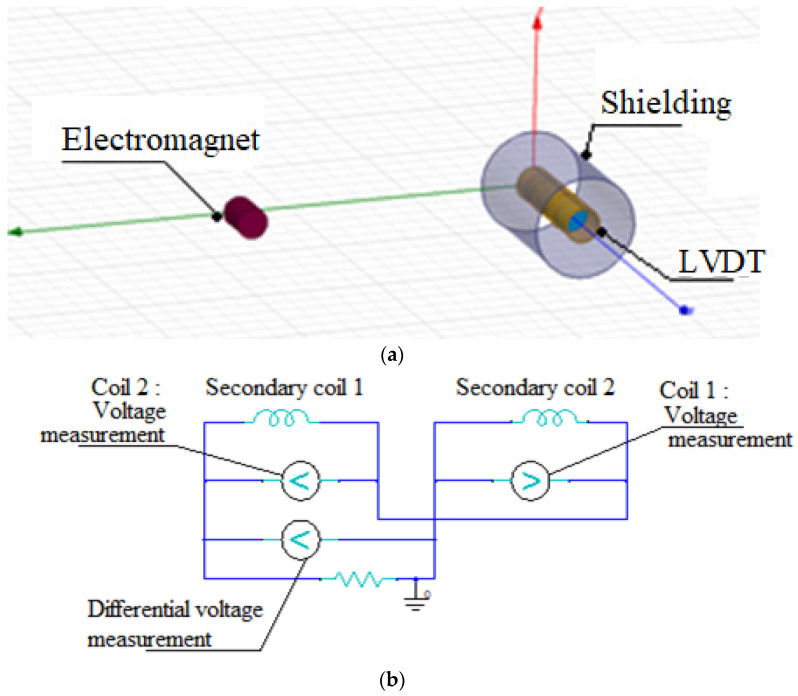
(**a**) FEA model of LVDT and electromagnet. (**b**) Electrical circuit in ANSYS Maxwell.

**Figure 18 materials-15-02630-f018:**
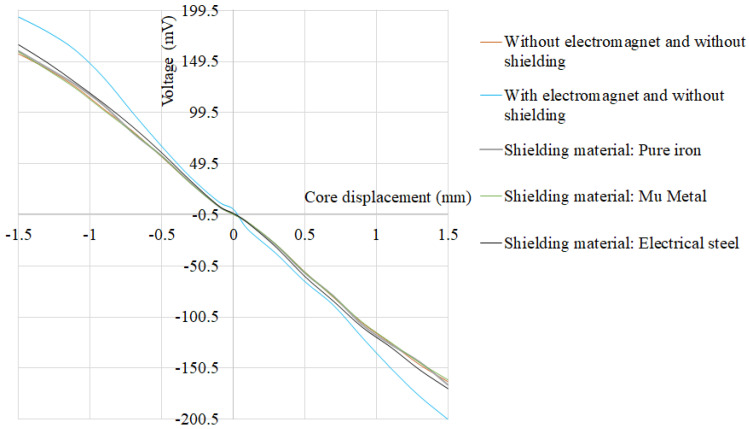
Secondary coil differential voltage along the displacement of the LVDT core.

**Table 1 materials-15-02630-t001:** Details of Electro-deposition process.

Cathode Electrode Details	Stainless Steel Plates
Dimension of the electrolyte bath	900 mm × 800 mm × 1200 mm
Chemicals used in the electrolyte solution	Iron chloride (320 kg), ammonium chloride (80 kg) and 400 Litres of water
Current density	16 A/m^2^
pH value range	5.00–5.42
Temperature of electrolyte	48–50 °C
Washing cycle duration	4.5 days
Deposition thickness	2.5–3 mm
Details of DC power source used	120 V, 600 Amp, Make: Kraft Powercon India Ltd., Pune, India

**Table 2 materials-15-02630-t002:** Chemical composition of Specimen.

Element	Ppm Level
Carbon	6.6–7.3
Sulphur	7.6–8.9
Nitrogen	4.4–5.1
Oxygen	105–121
Iron	Balance

**Table 3 materials-15-02630-t003:** Rolling mill parameter details.

Power Rating	10 HP
Mill size	178 cm × 76 cm × 158 cm
Separating force	1,620,000 lbs maximum
Tension	8000 lbs maximum
Work roll diameter	6.0 inches to 10.0 inches
Electric motor speed	1750 rpm
Roll speed	500 rpm (for initial 3 passes), 520 rpm (for remaining passes)
Number of passes to achieve 0.3 mm thickness	8–9
Material of the rolls used for rolling operation	Tungsten carbide
Mode of adjusting the rolling thickness	Hydraulically assisted

**Table 4 materials-15-02630-t004:** Changes in physical properties of the cold rolled iron strips.

Property	Before Cold Rolling	After Cold Rolling and Annealing
Density (gm/cc)	7.49	7.71
Thickness (mm)	3 mm	0.3–0.1 mm
Hardness (HV)	78.033	145.700

**Table 5 materials-15-02630-t005:** Magnetic properties of pure iron and electrical steel.

Property	
Remanent flux density	1.14 T
Coercivity	52 A/m
Maximum flux density	1.74 T
Maximum relative magnetic permeability	7818

**Table 6 materials-15-02630-t006:** Comparison of core loss reported for pure iron in the earlier works.

Reference	Core Loss
This work	8.1 W/kg at 50 Hz, with 0.16 mm-thick strip
[36]	4.6 W/kg at 50 Hz, with 0.1–1.0 mm-thick strip
[37]	21 W/kg at 50 Hz, with 0.30 mm-thick strip

**Table 7 materials-15-02630-t007:** Details of FEA for static and high frequency analysis.

Component	Ansys Element	Properties Defined
Air	Static analysis: Plane 233 Harmonic analysis: Plane13 Boundary: Infin 110	Relative permeability
Copper conductor	Static analysis: Plane 233 Harmonic analysis: Plane13	Relative permeability Electrical resistivity
Shielding material	Static analysis: Plane 233 Harmonic analysis: Plane13	B-H curve (determined in Section 3)
Material for eddy currents cancellation	Harmonic analysis: Plane 13	Relative permeability Electrical resistivity

**Table 8 materials-15-02630-t008:** Effect of plastic deformation on maximum relative permeability of pure iron.

Specimen Number	Strain Induced in the Pure Iron Strip	Maximum Relative Permeability as Measured with Two-Winding Method
1	0.00	6498
2	0.004	6498
3	0.01	5068

**Table 9 materials-15-02630-t009:** Details of miniature LVDT.

Measurement Range	3.0 mm (Total Range)
Number of turns in primary and secondary winding, respectively	60 and 80
Diameter of the copper wire in primary and secondary coils	0.4 mm
Voltage at the primary coils	3.0 V
Operating frequency	50 Hz

**Table 10 materials-15-02630-t010:** Properties defined in ANSYS.

Sr. No.	Name of the Component	Material	Properties Defined in ANSYS
1	Primary winding	copper	Number of turns, voltage, current and dimensions
2	Secondary winding	copper	Number of turns, and dimensions
3	Core	Iron	B-H curve of Pure iron from ANSYS library
4	Electromagnet	copper	Number of turns, voltage, current and dimensions
5	Magnetic shielding material	Pure iron	B-H curve determined from magnetic characterization as per IEC60404-04 for the pure iron manufactured in the present study (Initial magnetization curve)
6	Air	Air	Relative permeability

## Data Availability

Not Applicable.

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
