# Peer review of "Manufacturing of Pure Iron by Cold Rolling and Investigation for Application in Magnetic Flux Shielding"

_materials, 2022, doi:10.3390/ma15072630_

Round 1

Reviewer 1 Report

In the manuscript, the authors reported a method to manufacture iron foils by using electro-deposition and cold rolling along with intermediate annealing in a controlled environment. The magnetic and mechanical properties of the produced materials were investigated by different methods. Finite element analysis was also conducted to study the shielding performance of their material with that of the other candidate materials. For some aspects, many detailed information has been givens, such as some equipment and sample photos, different property characterization, and FEA simulation. However, there are still many shortcoming points in current manuscript.

  • The general organization of the manuscript must be improved. In introduction, more recent works concerning this topic should be critically reviewed. Many experimental details could be put in experimental part. Some key parameters about fabrication procedure and characterization are lacked.
  • There are many typos could be corrected in the manuscript.
  • The image quality should be improved, in particular, high resolution SEM images should be provided for microstructure information.
  • The authors underlined their method is novel and economic than current solution. In my opinion, their reported fabrication procedure was quite low and energy cost. And the electro-deposition and rolling method is also a very common way to produce metal foils. Moreover, the final performance of the materials obtained is not excellent. Please comment on these points.

Reviewer 2 Report

The authors present a study of pure iron manufactured by electro-deposition and cold rolling. The magnetic properties are evaluated and the shielding performance is calculated based on them. The purpose of the study is well addressed in the introduction and the article could perhaps be of some value regarding the obtention of cheaper materials for magnetic shielding, but the paper has several important flaws that make me recommend the rejection for publication.

  1. The document resembles more a technical report than a scientific publication. The quality of the presentation is poor, lacking important experimental details and focus. There is an excessive number of pictures, some even repeated (Figure 3b is the same as Figure 5). Pictures of commercial equipment may be omitted. At the same time, experimental details are lacking while less relevant information is provided in tables. There are numbering errors in the paragraphs and the structure of the paper is puzzling. The division into magnetic properties, finite elements analysis, and again reprocessing and magnetic properties is not very logical. It results difficult for the reader to follow the experimental results, analysis and discussion.
  2. The main point for magnetic shielding is to have high magnetic permeability at the magnetic field of the application. The magnetic permeability of a material varies with the applied magnetic field, it is a function, not a constant value. The value of the relative magnetic permeability is always given for a specific value of the magnetic field. This concept is wrong throughout the article, where the permeability is used as a constant, dissociated from the magnetic field strength.
  3. The main magnetic properties are obtained in Figure 9 and Table 5 by means of a VSM (vibrating Sample magnetometer). This is an open circuit magnetic measurement method which is not suitable for the determination of magnetic permeability due to demagnetizing effects on the sample specimen. The slope of the B-H curve for high permeability materials is inversely proportional to the demagnetizing factor, which is a geometric parameter of the sample. Therefore, all values ​​thus obtained are incorrect, unless otherwise determined. This problem invalidates the results presented in the paper. Furthermore, it is clearly seen that the values of coercivity of 80 A/m and 70 A/m do not match the width difference of the hysteresis loop. The same for the maximum flux density. In addition, the obtained value of 62000 for hydrogen annealed pure iron strips seems unrealistic.

Other minor mistakes:

- References should be reviewed, for example reference 12 does not correspond to what is detailed in the text.

- The residual magnetic flux density value of 39.2 uT must be referred to the value without shielding. (page 4).

- On page 4, the ‘Epstein’ method (written as abstain). Although this method is not discussed in section 3.  I assume this method refers to the IEC 60404-2 method of measurement of the magnetic properties of electrical steel strip and sheet by means of an Epstein frame, which is not used.

-Table2. How is the chemical composition determined? What method of analysis has been used?

- Cold rolling and annealing is section 2.2

-Table 4, what are the error bars.

-FEA paragraph is section 4.0

Reviewer 3 Report

This is an interesting study and the authors have collected a unique dataset using cutting-edge methodology on alternative magnetic shielding methods using pure Irons. The paper is generally well written and structured. Authors have covered both structural and magnetic property studies of pure Iron to establish their successful growth mechanism. The simulative process to demonstrate the shielding effectiveness is really a good asset of this manuscript. However, in my opinion, I think the manuscript would be complete if the authors would discuss a bit more about the low-frequency shielding process.

Author Response

According to the comments from Hon. Reviewer, low frequency magnetic shielding analysis at 50 Hz has been incorporated in the revised manuscript.

Reviewer 4 Report

1. The article has a high volume. The introduction contains 3.5 pages of text, which is not much justified. References to papers are presented without describing the main results - is there any sense in such a listing?
2. The photographs of EDS analysis 6.2, 7.2, 8.2 show the spectrum in the range from 0 to 2.5 keV. This is not enough to form an opinion about the complete chemical composition of the material.
3. Conclusion that "The manufacturing process presented in the this study is relatively inexpensive in comparison to that of the existing methods that involve hot rolling, die casting and controlled environment melting." is unreasonable, because The paper describes rather time-consuming operations of electro-deposition, rolling, and annealing. Casting and machining operations were related to the raw material, which was described in 2.1. In all places it is necessary to correct the economic parameter of superiority over standard methods of obtaining.
4. The work uses the operation of hydrogenation to reduce the proportion of oxygen, but does not talk about the mechanical properties of such a material. This is known to lead to brittleness.
5. In the conclusion section, it is necessary to present the main results point by point.

Reviewer 5 Report

  1. The present manuscript is too long. The quality of Abstract, Introduction, Results and discussion and Conclusion sections are need further improve. The figures have poor quality. Thus, I strongly recommend the authors to put some comments, figures and tables to add in the supporting information to have a better understanding for readers.
  2. Materials is paying more and more attention to the quality of articles, especially originality and innovation. The authors should state the innovation of the article more clearly and fully in the Abstract, Introduction, Results and discussion and Conclusion sections.
  3. The novelty/achievements of the work should be highlighted.
  4. The Figures are unaesthetic. I strongly recommend the authors to check the all figures and make beautiful.
  5. The authors should make a comparison with the results in recent publications.
  6. In Results and discussion section, the analysis is unclear and do not cite related references to further in-depth present the innovativeness and significance in this work.
  7. Finally, authors are also requested to review the manuscript to correct language related problems at several places.

Round 2

Reviewer 1 Report

A significant improvement has been mde by authors on answering the reviewer comments and the manuscript.  

Author Response

An attempt has been made in the revised manuscript to improve presentation of results as follows;

1. In context to the magnetic characterization which is important for the present study, VSM method has been replaced in the revised manuscript with methods specified in IEC 60404-4 standard, which is most appropriate for the present study with high permeable material.

2. Conclusions regarding purity of the pure iron, advantages of the proposed manufacturing method and applications for magnetic shielding is included in the revised manuscript (pls. refer page no. 28, lines 2-8 from top).

Lastly the entire manuscript has been reviewed very carefully for English language and suitable changes have been incorporated.

Reviewer 2 Report

See document. 

Reviewer 5 Report

Although the authors answered some of the questions in the revised manuscript, moderate revision of several points are required before the paper can be published.

  1. In Introduction section, the authors mentioned traditional approaches, electro-deposition and cold rolling approach. In fact, the electrolytic method also has potential application. The authors can give some comments in the Introduction section about the difference. I recommend the authors to cite some references, such as Rare Metals. 2021, 40(8): 2307-2312.
  2. Finally, the language of the manuscript is still required further improvement.
